# Integrative Metallomics Studies of Toxic Metal(loid) Substances at the Blood Plasma–Red Blood Cell–Organ/Tumor Nexus

**Maryam Doroudian and Jürgen Gailer ***

Department of Chemistry, 2500 University Drive NW, University of Calgary, Calgary, AB T2N 1N4, Canada
* Correspondence: jgailer@ucalgary.ca

**Abstract:** Globally, an estimated 9 million deaths per year are caused by human exposure to environmental pollutants, including toxic metal(loid) species. Since pollution is underestimated in calculations of the global burden of disease, the actual number of pollution-related deaths per year is likely to be substantially greater. Conversely, anticancer metallodrugs are deliberately administered to cancer patients, but their often dose-limiting severe adverse side-effects necessitate the urgent development of more effective metallodrugs that offer fewer off-target effects. What these seemingly unrelated events have in common is our limited understanding of what happens when each of these toxic metal(loid) substances enter the human bloodstream. However, the bioinorganic chemistry that unfolds at the plasma/red blood cell interface is directly implicated in mediating organ/tumor damage and, therefore, is of immediate toxicological and pharmacological relevance. This perspective will provide a brief synopsis of the bioinorganic chemistry of $As^{III}$, $Cd^{2+}$, $Hg^{2+}$, $CH_3Hg^+$ and the anticancer metallodrug cisplatin in the bloodstream. Probing these processes at near-physiological conditions and integrating the results with biochemical events within organs and/or tumors has the potential to causally link chronic human exposure to toxic metal(loid) species with disease etiology *and* to translate more novel anticancer metal complexes to clinical studies, which will significantly improve human health in the 21st century.

**Keywords:** toxic metal(loid)s; chronic exposure; mechanism of toxicity; metallodrugs; stability in plasma; side effects; bloodstream; bioinorganic chemistry

## 1. Introduction

Ever since the inception of life some 3.8 billion years ago—possibly in the vicinity of deep-sea hydrothermal vents [1]—it has been continually exposed to background concentrations of inherently toxic elements released from the earth's crust into the biosphere by natural processes, including volcanism and chemical weathering. Therefore, all organisms evolved in the presence of potentially toxic metal(loid) species and had to adapt to background concentrations over millions of years [2]. A momentous event in the 1760-1780s, however, changed the organism–earth relationship forever: the industrial revolution. This represents the onset of the production of consumer products at an ever-increasing scale, which required unprecedented amounts of energy (i.e., fossil fuels) and building blocks (i.e., chemical elements). The associated advent of the mining/metallurgy industry resulted in the increased emission of toxic metal(loid) species into the environment in many parts of the world [3,4], which has dramatically affected ecosystems [5] and, in turn, inevitably exposed certain human populations [6].

A conceptually related event which involved the interaction of an entirely different type of toxic metalloid compound with humans took place in 1911: the birth of chemotherapy by Paul Ehrlich. While his synthesis of the arsenic-containing drug Salvarsan contributed to humankind's exploration of the available 'chemical space' [7], its administration to patients who suffered from syphilis—a bacterial infection caused by *Treponema pallidum*—ushered in a fundamentally new therapeutic approach to treat human diseases by exploiting the concept of 'selective toxicity' [8,9].

What conceptually relates the exposure of humans to toxic metal(loid) species with the administration of an arsenic-containing drug to patients is the fact that both of these toxic substances infiltrate the systemic blood circulation, which was discovered by William Harvey [10]. His recognition that the bloodstream effectively represents a conveyor belt which supplies human organs with water, life-sustaining oxygen and nutrients (essential elements, vitamins, carbohydrates and lipids) to maintain human health makes this biological fluid one of the first sites where adverse interactions between toxic metal(loid) species/cytotoxic metallodrugs and constituents of the bloodstream unfold (Figure 1) [11,12]. The toxicological and pharmacological relevance of these interactions with constituents of the bloodstream is attributed to the simple fact that they collectively constitute a 'filter' which fundamentally determines which toxic-metal-containing species or metabolites thereof will impinge on organs to cause either a detrimental effect, in the case of toxic metal(loid) species, and/or a desirable pharmacological effect in the case of anticancer metallodrugs on a malignant tumor.

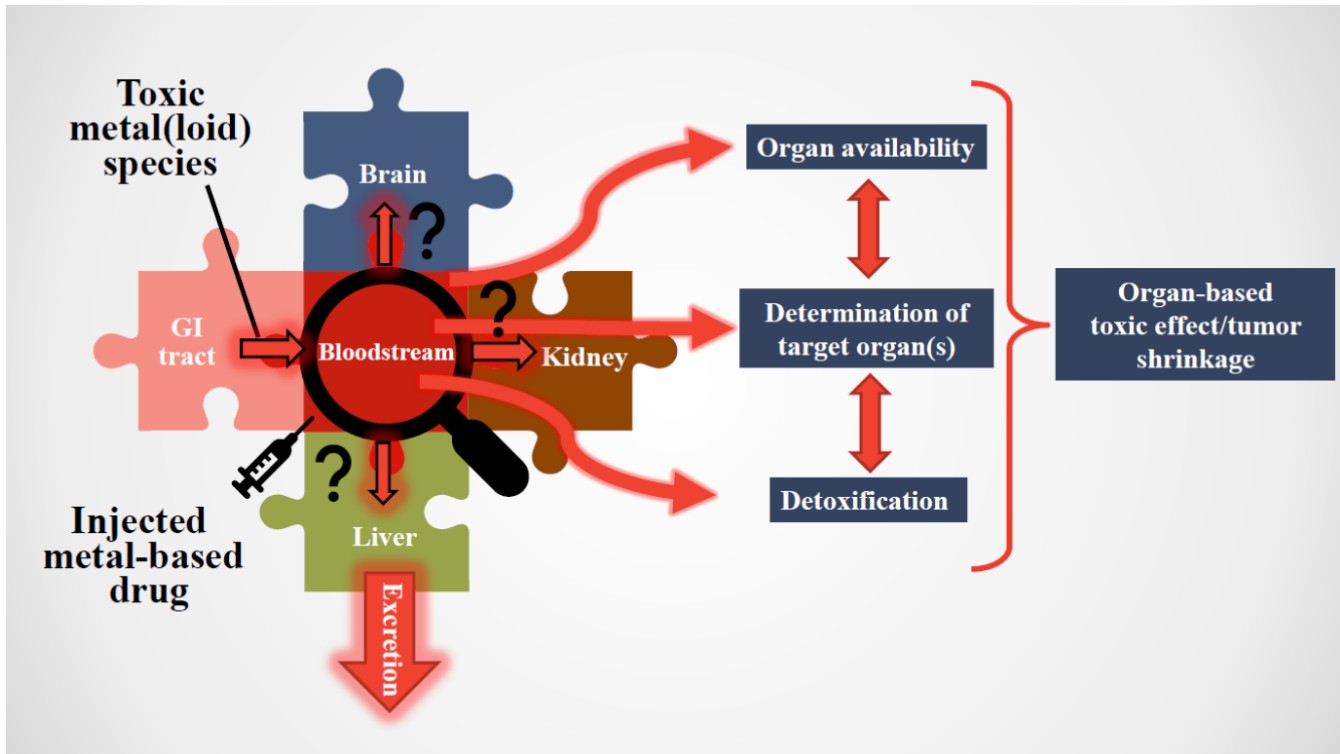

**Figure 1.** The central role held by the bioinorganic chemistry of toxic metal(loid) species and metallodrugs in the bloodstream in terms of linking environmental exposure to adverse health effects/organ-based diseases and in developing better drugs to treat human diseases, such as cancer (Figure modified from [13]).

Considerable progress has been made to conceptually relate toxic metal(loid) species and metallodrugs—which will from now on be referred to as toxic metal(loid) substances—in biological systems to adverse or intended human health effects. It is, therefore, timely to provide the reader with a perspective as to why the bioinorganic chemistry of toxic metal(loid) substances, specifically in the bloodstream, has become a worthwhile research avenue. Since the available experimental techniques to measure metal-containing metabolites in biological fluids are not our primary focus, the interested reader is referred to relevant reviews [14–18]. Nevertheless, a few techniques that are particularly appropriate to solve relevant problems will be outlined. Then, bioinorganic mechanisms of toxic metal(loid) species in the bloodstream will be presented, and their conceptual link to adverse health effects will be discussed. This section will be followed by an analogous brief

summary of the biochemical fate of anticancer metallodrugs in the bloodstream. Finally, we underscore the importance of integrating processes involving toxic metal(loid) substances in the bloodstream with those that happen in organs in terms of understanding their impact at the whole-organism level, either negatively (toxicology) or advantageously (pharmacology).

## 2. Critical Importance of Studying Toxic Metal(loid) Substances in the Bloodstream (Footnote: From Now on We Use the Term 'Substances' When We Refer to Both Toxic Metal(loid) Species and Metallodrugs)

The man-made emission of toxic metal(loid) species into the environment has gradually increased since the onset of the Industrial Revolution, and perturbs the global biogeochemical element cycles of at least 11 elements today, with arsenic (As), cadmium (Cd) and mercury (Hg) species critically affecting air, water and food [19]. In fact, the input of Hg to ecosystems is estimated to have increased two- to five-fold during the industrial era [20]. These profound geochemical changes in the biosphere have prompted some scientists to refer to our age as the 'Anthropocene' [21]. Certain human populations, including children, are therefore unwittingly exposed to higher daily doses of persistent inorganic environmental pollutants through the ingestion of contaminated food [22–25], the inhalation of contaminated air [26–29] and/or dermal exposure to consumer products [30–32] than ever before. Unsurprisingly, pollution has become the world's largest environmental cause of disease and premature death, with an estimated ~9 million deaths per year [33] and is associated with staggering global healthcare costs [34]. Owing to the inherent persistence of As, Cd and Hg species and their binding to hematite and humic acids in soils [35], their concentrations in ecosystems will either remain constant—potentially for millennia [36,37]—or even gradually increase over time [38,39]. Due to the continued consumption of large quantities of Hg [40], the global-warming-induced re-mobilization of certain metal(loid) species [41] and the ongoing contamination of the food chain with these inorganic pollutants [22,25,42,43], the concomitant exposure of human populations represents a global health problem [44–46]. Owing to the severe health effects associated with human exposure to comparatively small daily doses of toxic metal(loid) species (up to 260 μg/day), the toxicology of metals has received considerable research attention [6,47], which has in turn resulted in the implementation of guidelines for maximum permissible concentrations in drinking water [48], food [22] and air [49] in many countries. As a consequence of additional human exposure to these inorganic pollutants through consumer products [30], personal care products [32], nanomaterials [50] and the global-warming-induced increase in using wastewater for food irrigation [44], an emerging challenge is the establishment of a new paradigm to assess how a lifetime of exposure affects the risk of developing chronic diseases [51]. In this context, the elucidation of the biomolecular mechanisms of toxicity associated with chronic human exposure to toxic metal(loid) species [52,53] and causally linking these mechanisms with the etiology of human illnesses that do not have a genetic origin [54] remain perhaps the two most pertinent knowledge gaps.

The development of chemotherapy to treat human diseases other than bacterial infections was—somewhat counterintuitively—accelerated during World Wars I and II, when certain warfare agents were used for the first time. In particular, the observation that soldiers' exposure to nitrogen mustards specifically targeted bone marrow cells prompted scientists to investigate if this effect may be harnessed to selectively target malignant cells. By the mid-1960s, the anticancer metallodrug cisplatin was serendipitously discovered [55] and FDA-approved in 1978. Despite cisplatin being a 'shotgun' cytotoxin (it does not differentiate between cancer and healthy cells), which is intrinsically associated with severe and dose-limiting side-effects [56], close to 50% of all cancer patients worldwide are today treated with cisplatin, carboplatin and/or oxaliplatin, either by itself or in combination with other anticancer drugs [12]. The success story of cisplatin triggered intense research efforts to develop metallodrugs for use as photochemotherapeutic drugs, antiviral drugs, antiarthritic drugs, antidiabetic drugs, drugs for the treatment of cardiovascular and gastrointestinal disorders as well as psychotropics [57], but overall metallodrugs remain a tiny

minority of all medicinal drugs that are currently on the market [58,59]. The considerable burden of cancer on the world economy [60], however, necessitates the urgent development of more effective anticancer drugs that offer higher selectivity and, thus, fewer side-effects to improve the quality of life of patients during and after cancer treatment. Even though promising novel anticancer metal complexes are being developed [9,61], advancing more of these to preclinical/clinical studies remains a major problem [12]. This problem is attributed to the fact that insufficient attention is being directed to assess the effect of pharmacologically relevant doses of novel anticancer metal complexes on (a) the integrity of red blood cells (RBCs) [62], (b) their stability in blood plasma [12,63] and (c) their selectivity toward cancer cells versus healthy cells [64].

## 3. General Considerations Pertaining to Interactions of Toxic Metal(loid) Substances in Humans

With regard to the infiltration of the bloodstream by a toxic metal(loid) species, a reasonable question to ask is how an exceedingly small daily dose (e.g., 200 µg of inorganic As/day will eventually result in cancer [65]) can reach target organs. Interactions of toxic metal(loid) species with some bloodstream constituents are 'good', as their initial binding/sequestration (e.g., binding to human serum albumin or uptake into RBCs) will delay/preclude it from reaching a target organ. Conversely, a metallodrug that is intravenously administered is intended to reach the malignant tumor intact in order to cause maximal damage. Hence, any interaction(s) of an anticancer metallodrug with bloodstream constituents (e.g., a partial decomposition followed by plasma protein binding) is/are 'bad', as less will reach the tumor tissue.

It is important to point out that the daily human exposure to toxic metal(loid) species by inhalation and ingestion is comparatively lower (µg/day) than that of anticancer metallodrugs, which are intravenously administered (mg/day). It is, therefore, not entirely surprising that an extensive lag-time can exist between the onset of chronic human exposure to toxic metal(loid) species and the start of overt adverse health effects [52], while overt signs of toxicity manifest themselves typically within hours or days after the intravenous administration of patients with anticancer metallodrugs [56].

The challenge that pertains to both types of these toxic metal(loid) substances is the need to establish the entire sequence of bioinorganic interactions that occur at the plasma/RBC interface and to integrate them with processes that unfold in organs/tumors (Figure 2). For toxic metal(loid) species, this means revealing the biomolecular mechanisms causing organ damage; for metallodrugs, it also implies screening novel molecular structures which preferentially target the tumor cells, while leaving healthy organs unscathed. Ironically, the effect of these toxic metal(loid) substances on human health are conceptually intertwined in an interesting manner, since specific toxic metal(loid) species are established carcinogens (e.g., inorganic As and Cd [66,67]), while cancer patients are being treated with highly cytotoxic metallodrugs (e.g., cisplatin) to achieve tumor shrinkage/remission that are often associated with severe side-effects.

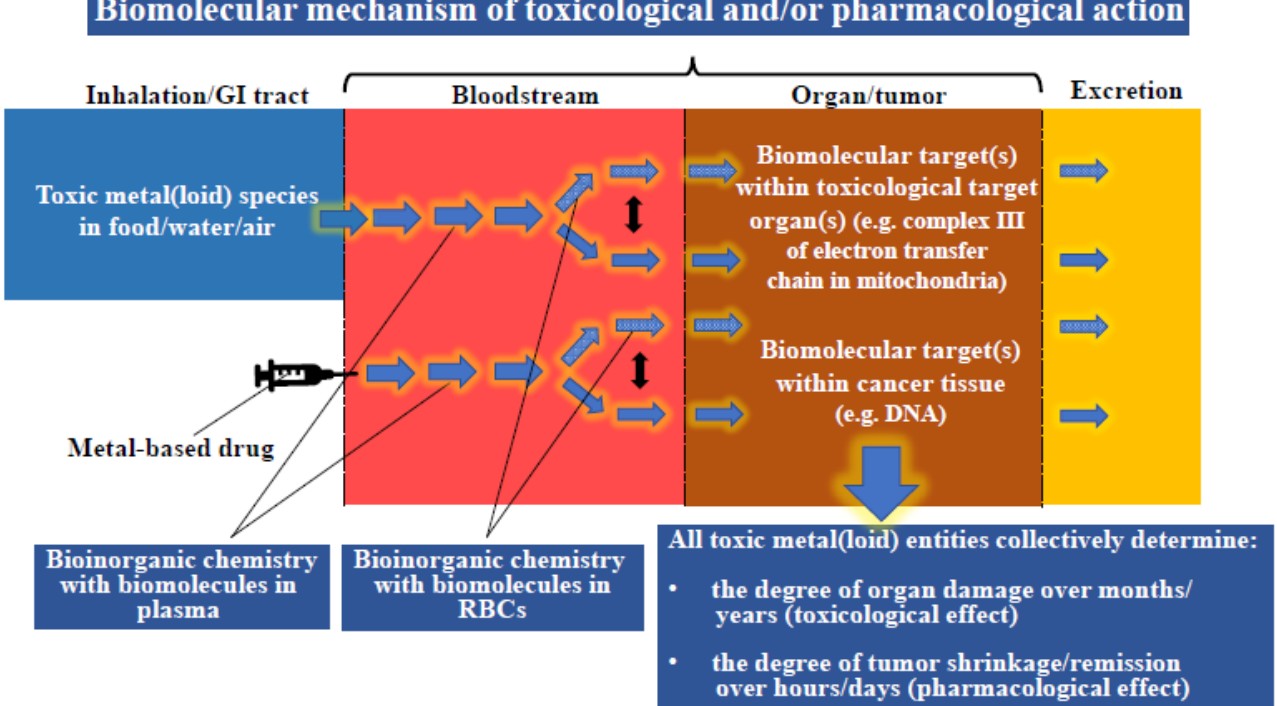

**Figure 2.** Conceptual relationship of bioinorganic processes in the bloodstream following (**a**) the chronic exposure of humans to a toxic metal(loid) species (toxicological effect) and (**b**) the intravenous administration by cancer patients of an anticancer metallodrug (pharmacological effect).

## 4. Bioinorganic Chemistry of Toxic Metal(loid) Substances in the Bloodstream In Vitro/In Vivo

One of the main difficulties of studying chemical reactions that unfold in the bloodstream is its biological complexity, which is directly related to its orchestration of the continual exchange of nutrients absorbed from the GI tract to organs and the eventual excretion of waste products via the kidneys. The bloodstream should not, however, be viewed as a seamless pipe, since it contains up to 10,000 plasma proteins [68], >400 SMW metabolites [69] and ~1600 RBC cytosolic proteins [70] which can engage in a variety of potential chemical reactions with toxic metal(loid) substances. The enormous number of potential binding sites on plasma proteins/SMW metabolites, for example, can result in their reversible vs. irreversible binding [71], while the partial or complete sequestration of toxic metal(loid) substances within RBCs would preclude their influx to target organs. Given the reducing conditions within RBCs, redox reactions of toxic metal(loid) substances within RBCs must also be considered. The intrinsic biological complexity of plasma and RBC cytosol can be overcome by using metallomics tools, which refers to hyphenated techniques that are comprised of a separation method (e.g., HPLC, 2D-PAGE) coupled to an element-specific detection technique [14–16,72]. The application of metallomics tools inherently reduces the analytical separation problem significantly, as the number of endogenous metal species within any given biological fluid (e.g., all endogenous metalloproteins) represents a sub-proteome of the proteome (i.e., all proteins within a biological fluid).

Detecting complexes containing a chemical bond between a toxic and an essential element, for example, in blood plasma, would reveal which essential trace element is targeted by a particular toxic metal(loid) species [73]. Since blood is comprised of about 50% of RBCs, it is also of importance to consider the binding of toxic metal(loid) species to the lipid bilayer membrane of intact RBCs [74] and the cytosolic proteins within RBCs [75]. In terms of anticancer metallodrugs, their metabolism in the bloodstream can lead to the formation of hydrolysis products, which can then bind to plasma proteins, such as human serum albumin (HSA). It is therefore desirable to analyze blood plasma for all

parent metal-containing metabolites, which has been successfully attained for cisplatin and carboplatin [76].

In addition to analyzing plasma and RBC cytosol, one should also heed the advice from R.J.P. Williams that 'living organisms cannot be understood by studying extracted (dead) molecules. We have to study flow systems' [77]. Accordingly, conducting in vivo experiments using animal models is invaluable as it can provide an important starting point for further research. For example, it may be possible to identify a particular complex which contains a chemical bond between a toxic and an essential element [78,79] which can then initiate studies to establish the entire mechanism of chronic toxicity/pharmacological action within the bloodstream–organ system (Figure 2). To this end, our lab has demonstrated that the metabolism of $As^{III}$ is fundamentally tied to that of the essential trace element selenium. After the intravenous injection of rabbits with $As^{III}$ and $Se^{IV}$, the analysis of rabbit bile revealed an As:Se molar ratio of 1:1, which implied the in vivo formation and excretion of an As-Se compound from the liver. The latter compound was structurally characterized as the seleno-bis(S-glutathionyl) arsinium ion $[(GS)_2AsSe^-]$ [78], which is formed intracellularly following Equation 1.

$$As(OH)_3 + HSeO_3^- + 8GSH \rightarrow (GS)_2AsSe^- + 3GSSG + 6H_2O \tag{1}$$

Notably, the results from this in vivo experiment were crucial to design subsequent investigations, which revealed that $(GS)_2AsSe^-$ is formed in the bloodstream [80] (Figure 3). More recent experiments revealed that this metabolite is present in the liver, the gall bladder and the GI tract [81], and experiments with rabbits revealed that this $As^{III}$-$Se^{IV}$ antagonism is of direct environmental relevance [82]. Similarly, the formation and structural characterization of a Hg- and Se-containing detoxification product that is rapidly formed in rabbit blood was first observed in vivo [79]. The formation of the detoxification product $(HgSe)_{100}SelP$ (Figure 3) could not have been observed in blood plasma or RBCs alone, as its mechanism of formation involves the reduction of $Se^{IV}$ to $HSe^-$ within RBCs followed by its subsequent excretion into plasma where it then reacts with $Hg^{2+}$. These few examples underscore the vital role that in vivo studies play in the context of discovering novel metabolites of toxicological importance.

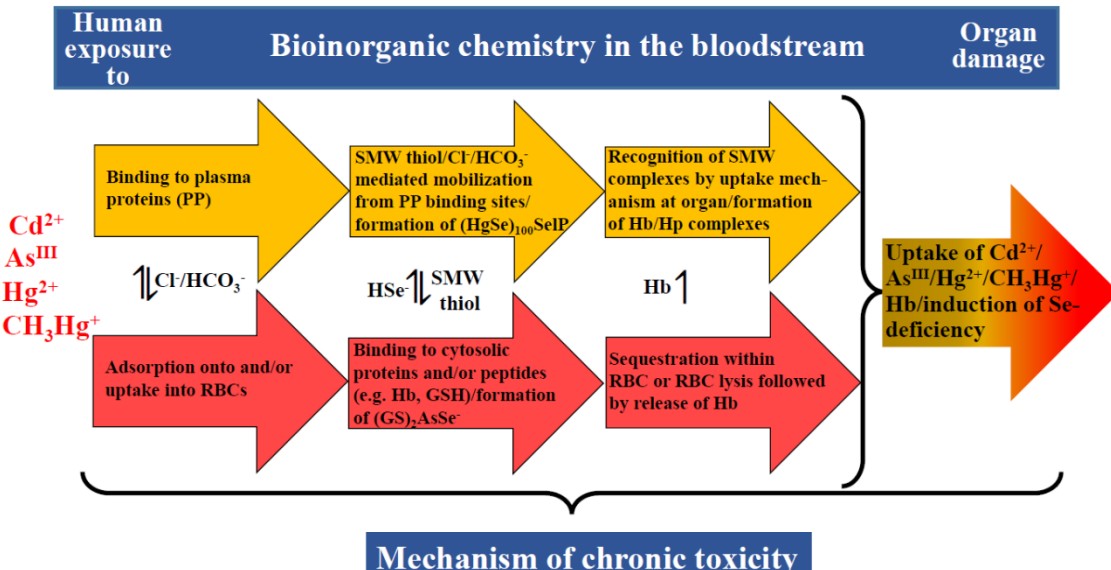

**Figure 3.** Bioinorganic chemistry of toxic metal(loid) species that unfold in plasma (yellow) and red blood cells (red) are critical to establish their mechanism of chronic toxicity, which determines the degree of organ damage over weeks/months/years. Abbreviations: red blood cells (RBC), small molecular weight (SMW), hemoglobin (Hb), selenoprotein P (SelP), haptoglobin (Hp), glutathione (GSH).

### 5. Toxic Metal(loid) Species at the Plasma–RBC–Organ Nexus

From a conceptual point of view, the 'missing link' in the toxicology of metal(loid) species ultimately lies in causally linking molecular data with human health effects [83]. At face value, this problem translates to an enormous transdisciplinary chemistry–biology problem because 'molecular data' refers to toxicologically relevant interactions of any given toxic metal(loid) species with biomolecules in the bloodstream [84] and in target organs [85] which then collectively result in a specific adverse health effect at the whole organism level (e.g., kidney damage). Understanding the associated exposure–response relationship is critically dependent on predicting how much organ damage a toxic metal(loid) species absorbed into the bloodstream ($As^{III}$, $Cd^{2+}$, $Hg^{2+}$ and $CH_3Hg^+$) can ultimately inflict. This goal requires one to unravel toxicologically relevant bioinorganic events in the bloodstream, which can involve chemistry of the toxic metal(loid) species in plasma (Figure 3, yellow arrows) [84] and/or in RBCs (Figure 3, red arrows) [75]. While these events can be studied separately, one must integrate the results to establish the metabolism of the toxic metal(loid) species at the plasma–RBC interface (see the bidirectional arrows between the yellow and the red arrows in Figure 3), which ultimately determines how much of the initial dose will be able to engage in organ damage 'downstream' [86]. The dynamic bioinorganic events that unfold at the plasma–RBC–organ nexus are, in all likelihood, also responsible for the considerable lag-phase that can exist between the onset of the chronic exposure of an organism to toxic metal(loid) species and the time when adverse organ-based toxic effects manifest themselves [52,87].

From a mechanistic point of view, chronic human exposure to toxic metal(loid) species can involve three bioinorganic processes that happen in the bloodstream, which may considerably reduce the fraction of the toxic metal(loid) species that is not 'detoxified' therein, and therefore, are of toxicological relevance. One important process is the sequestration of a toxic metal(loid) species within RBCs (e.g., $CH_3Hg^+$ [88], Cd [89]) or the formation of non-toxic complexes with essential elements, such as selenium either in blood plasma [73] or RBCs [90] (Figure 3). Another process that needs to be considered is a toxic-metal(loid)-species-mediated lysis of RBCs, which releases Hb to plasma where it tightly binds to the plasma protein haptoglobin (Hp). Since the binding capacity of the latter is limited by its plasma concentration, however, the release of too much Hb can result in 'free' Hb in plasma which can then cause severe kidney damage [91]. Yet another process by which a toxic metal(loid) species may contribute to organ damage is the gradual induction of a trace-element deficiency based on events that unfold in the bloodstream–organ system [73,81,86].

Last but not least, one needs to consider a toxic-metal(loid)-species-induced adverse effect on the assembly of RBCs in the bone marrow, which may adversely affect the cytosolic concentrations of metalloproteins in the RBCs that are released into the blood circulation [e.g., a decreased hemoglobin (Hb) concentration therein corresponds to anemia], or may result in a decreased lifetime of RBCs in the blood circulation [92]. The exposure of mammals to a toxic metal(loid) species can also result in the hypoproduction of the hormone erythropoietin, which stimulates the production of RBCs and would, therefore, result in another form of anemia [93]. Taken together, all of these individual bioinorganic mechanisms collectively contribute to organ damage over weeks, months and possibly years depending on the dose of the toxic metal(loid) species.

From a practical view, gaining insight into the bioinorganic chemistry of toxic metal(loid)s in the bloodstream is hampered by its intrinsic biological complexity. The seemingly simple task of identifying those plasma proteins that specifically bind any given toxic metal(loid) species translates—owing to the presence of thousands of plasma proteins—into a considerable separation problem, but can be dramatically simplified by using so-called metallomics tools [13,16]. The simultaneous detection capability of some metallomics tools allows one to use the ~10 endogenous plasma/serum metalloproteins that contain transition metals (Cu, Fe and Zn) (Table 1) [94,95] and the ~4 metalloproteins that are present in RBC lysate (Table 2) as internal standards, which inherently represent molecular-weight markers and,

thus, validate an obtained analytical result. Furthermore, exogenous toxic metal(loid) species, such as $As^{III}$, $Cd^{2+}$, $Hg^{2+}$ and $CH_3Hg^+$ can also bind to the aforementioned metalloproteins and/or proteins in plasma and/or RBC cytosol. In blood plasma, for example, $Cd^{2+}$ has been shown to bind to serum albumin in rats [96]. Likewise, cisplatin-derived metabolites have been demonstrated to bind to HSA in human plasma [76], but the binding appears to be irreversible. In this context, the irreversible binding of a metallodrug to HSA renders the formed metallodrug–HSA complex ineffective for uptake into tumor cells [58], unless the tumor cells can able to achieve this [97] (e.g., by macropinocytosis). Furthermore, $Hg^{2+}$, $CH_3Hg^+$ and $CH_3CH_2Hg^+$ have each been demonstrated to bind to Hb in the matrix of RBC cytosol [75]. The formation of adducts between a toxic metal species and a metalloprotein can have important toxicological ramifications. For example, the binding of $CH_3CH_2Hg^+$ to Hb is associated with its conformational change, which significantly decreases its $O_2$ binding capacity [98].

**Table 1.** Characteristics of major metalloproteins in human plasma.

| Metal | Metalloprotein or Biomolecules which Contain Bound Metal(s) | Molecular Mass (kDa) | Number of Metal Atoms Bound per Protein | Reference |
|---|---|---|---|---|
| Fe | Ferritin | 450 | <4500 | [94] |
| | Transferrin | 79.9 | 1 | [94] |
| | Haptoglobin–Hemoglobin complex | 86–900 | 2 | [99] |
| Cu | Blood coagulation factor V | 330 | 1 | [94] |
| | Transcuprein | 270 | 0.5 | [94] |
| | Ceruloplasmin | 132 | 6 | [94] |
| | Albumin | 66 | 1 | [94] |
| | Extracellular Superoxide Dismutase | 165 | 4 | [94] |
| | Peptides and amino acids | <5 | - | [94] |
| Zn | $\alpha_2$ macroglobulin | 725 | 5 | [94] |
| | Albumin | 66 | 1 | [94] |
| | Extracellular Superoxide Dismutase | 165 | 4 | [94] |

**Table 2.** Characteristics of major metalloproteins in red blood cell cytosol.

| Metal | Metalloprotein | Molecular Mass (kDa) | Number of Metal Atoms Bound per Protein | Reference |
|---|---|---|---|---|
| Fe | Hemoglobin | 64.5 | 4 | [100] |
| | Catalase | 240 | 4 | [101] |
| Zn | Carbonic anhydrase | 30 | 1 | [102] |
| | Superoxide Dismutase | 32 | 1 | [103] |
| Cu | Superoxide Dismutase | 32 | 1 | [103] |

To date, the application of LC-based metallomics tools has provided important new insight into the bioinorganic chemistry of toxic metal(loid)s in the bloodstream (Table 3) by observing bi- and trimetallic complexes that contain essential and toxic metals [69], which contributed important insight into the biomolecular mechanisms that mediate organ damage (Figure 3) [86]. In terms of probing the interaction of toxic metal(loid) species with constituents of plasma, it is important to note that the latter contains about 400 small molecular weight (SMW) molecules and metabolites, including amino acids, peptides, fatty acids and nucleotides that are present at μM concentrations [104]. The variation in concentration of these SMW molecules in the bloodstream has been demonstrated to be,

in part, genetically determined [104]. These SMW molecules and metabolites are likely to play an important role in the distribution of toxic metal species to target organ tissues [105] (Figure 3). Direct experimental evidence in support of this comes from a recent study in which a metallomics tool was employed to demonstrate that homocysteine (hCys) is critically involved in the delivery of $CH_3Hg^+$ to L-type large neutral amino acid transporters (LATs) [106], which mediate the uptake of $CH_3Hg^+$-Cys complexes into the brain [84]. hCys, an intermediate metabolite formed by the de-methylation of methionine [107], may therefore be also involved in the translocation of other toxic metal(loid) species to target organs. It is possible that in patients with hyperhomocysteinemia— who exhibit elevated levels of hCys in blood plasma (e.g., >15 μM) which has been linked to the development of cardiovascular disease, stroke, and Alzheimer's Disease (AD) [107]—the metabolism of toxic metal(loid)s is significantly altered and may potentially exacerbate disease progression. Other SMW species that are also likely to be implicated in the organ uptake of toxic metal(loid) species are $Cl^-$ and $HCO_3^-$ [108], which may therefore be indirectly implicatedin neurodegenerative diseases [109].

## 6. Metallodrugs/Novel Metal(loid) Complexes at the Plasma–RBC–Organ–Tumor Nexus

Metallodrugs [57], including cisplatin—the oldest and best-known metallodrug [110]—are used as therapeutics for the treatment of various diseases; most prominently, cancer. However, the severe toxic side-effects of this Pt-based anticancer metallodrug prompted the development of other Pt-based drugs, such as oxaliplatin which kills cells by inducing ribosome biogenesis stress rather than the cisplatin-mediated DNA-damage response [111]. An alternative strategy to develop better metallodrugs is to embrace different metals altogether, which has resulted in the synthesis of anticancer active Cu complexes as well as ferrocene-functionalized Ru(II) arene complexes with interesting pharmacological properties [112,113]. Despite these encouraging advances, the translation of results from in vitro studies to therapeutic success using in vivo animal models, and eventually clinical studies, remains a major challenge. Conceptually there are two reasons why metallodrugs may be inherently better-suited to target disease-relevant proteins than some one-dimensional (1D, linear) or two-dimensional (2D, planar) organic molecules. Firstly, metallodrugs allow one to build a drug molecule that provides a wider range of geometries around a metal center in three dimensions (3D conformation) compared to 75% of linear and planar organic FDA-approved drug molecules, which offers the possibility of more-selectively inhibiting the active site of a target protein [114–117]. Secondly, certain metallodrugs can induce long-lasting anticancer immune responses [118], thus offering the potential to develop 'smart' metallodrugs [119]. The latter term refers to drugs that more-selectively target cancer cells, which is in accordance with the goal of achieving 'precision medicine' [120]. Despite these potential advantages and the fact that many novel structures are reported annually, only a negligibly small number of novel metal(loid) complexes enter clinical studies [112,117,121]. This undesirable situation is attributed to a) the common misperception that, similar to cisplatin [56], all metallodrugs are associated with severe toxic side-effects and b) the challenge of translating more metal(loid) complexes through systematic in vitro and in vivo studies to successful metallodrugs. The magnitude of the latter dilemma is illustrated by the fact that certain metal complexes exhibit potent cytotoxicity toward cancer cell lines in vitro, but show no antitumoral activity in vivo and may even display nephrotoxicity [122]. These facts hint at a transdisciplinary 'chemistry–medicine' problem of equal proportions to the aforementioned 'chemistry–biology' problem (see Section 5).

The fundamental problem that is associated with the assessment of novel metal entities—at least in our opinion—is that their interactions with constituents of the bloodstream are often not sufficiently considered [123]. We would like to illustrate this important bottleneck in the assessment of novel metal entities based on what is known about the fate of cisplatin in the human bloodstream (Figure 4). While cisplatin is an ideal candidate in this context because we know more about its biochemical fate in the bloodstream than any

other anticancer active metallodrug, we point out that this prototypical metallodrug lacks the desired tumor selectivity.

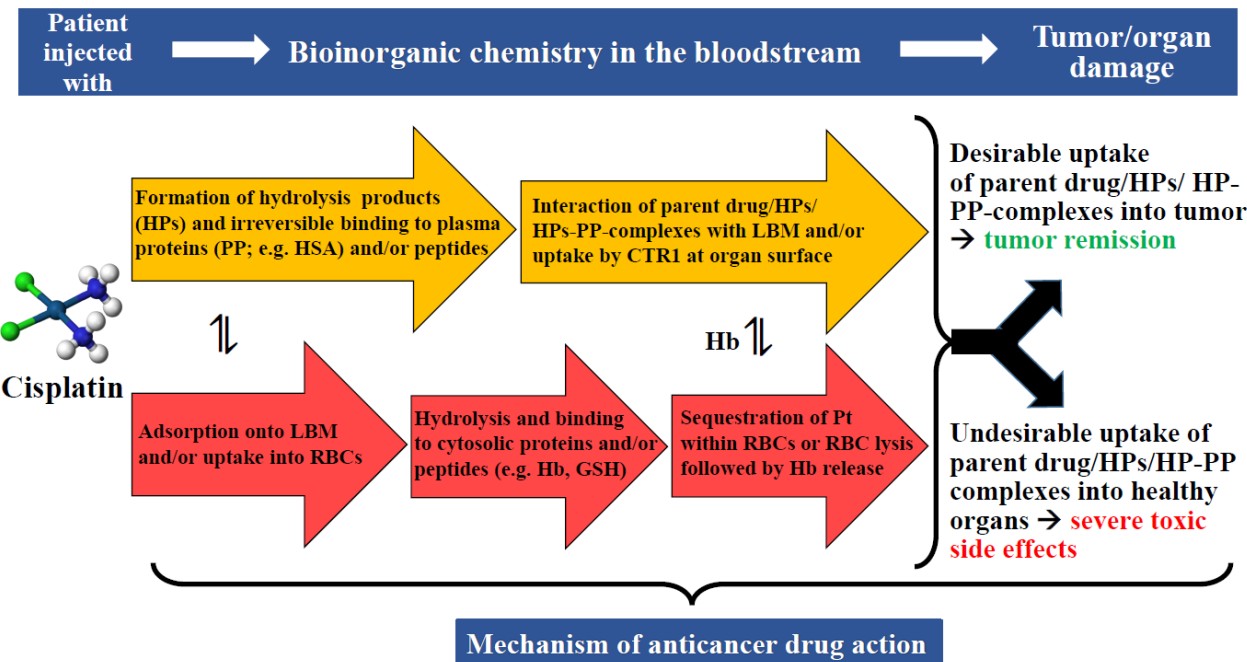

**Figure 4.** Bioinorganic chemistry of cytotoxic anticancer metallodrugs that unfold in plasma (yellow) and red blood cells (red) are critical to establish their mechanism of action, as they determine the degree of tumor damage/tumor shrinkage over days/weeks. Abbreviations: human serum albumin (HSA), red blood cells (RBC), hemoglobin (Hb), glutathione (GSH), lipid bilayer membrane (LBM), copper transporter 1 (CTR1).

## 6.1. Stability of Metallodrugs/Novel Metal(loid) Complexes in Plasma

A low temporal stability of a novel metal complex in plasma will dramatically reduce its probability of reaching the cancer tissue intact, and thus, increases the probability of severe toxic side-effects that are caused by the formed degradation products [120]. Compared to small organic molecules, this potential problem is much more troublesome for metallodrugs since a rupture of the bond that anchors the metal to its drug framework will compromise the integrity of the drug molecule, in turn adversely affecting not only the intended pharmacological effect and possibly also contributing to its severe adverse side-effects. Metallomics tools, which involve the hyphenation of a separation technique [e.g., capillary electrophoresis (CE), size-exclusion chromatography (SEC), high-/ultra-performance liquid chromatography (HPLC/UPLC)] with a metal-detector [e.g., inductively-coupled plasma mass spectrometry (ICP-MS), inductively coupled plasma atomic emission spectroscopy (ICP-AES)] can provide much-needed insight into the temporal stability of metallodrugs in serum [124].

The anticancer drug cisplatin, for example, undergoes several bioinorganic chemistry processes in blood plasma, including hydrolysis followed by the binding of the formed hydrolysis products to plasma proteins [76]. These dynamic bioinorganic processes gradually decrease the plasma concentration of the parent drug that is able to reach the intended tumor tissue intact, while the formed hydrolysis products are likely responsible for its severe side-effects (Figure 4) [76]. In this context, it is important to point out that the mere observation of a degradation product of a novel metal complex in blood plasma using a metallomics tool cannot provide any information about its toxicity, as this information has to be established using toxicological assays. The capability of metallomics tools to detect highly cytotoxic Pt species in plasma is reminiscent of their ability to identify potentially toxic metal species in human serum from environmentally exposed people [125] and to

determine natural variations in the isotopic composition of $CH_3Hg^+$ and $Hg^{2+}$ in fish tissue [126]. The application of SEC-ICP-AES has been particularly valuable to demonstrate that 70% of the bimetallic Ti- and Au-containing complex Titanocref remains intact in human plasma after incubation for 60 min at 37 °C [63], while the Ti- and Au-containing degradation products eluted bound to HSA. This comparatively novel metallomics tool has also been successfully applied to compare in vitro the metabolism of pharmacologically relevant doses of the FDA-approved platinum drugs cisplatin and carboplatin in human blood plasma [76]. The results revealed a comparatively faster hydrolysis of cisplatin followed by the binding of the formed hydrolysis products to plasma proteins, predominantly HSA, over a 24 h period compared to carboplatin. In addition, the hydrolysis products that formed from each platinum drug appeared to bind to the same plasma proteins. Importantly, these temporal changes in the metabolism of cisplatin/carboplatin provided an explanation for their considerably different toxic side-effects in patients [76] and the different biodistribution of these platinum drugs to organs after treatment.

Conceptually related to the application of SEC-ICP-AES to probe the stability of metallo-drugs and novel metal complexes in plasma, this metallomics tool has also allowed researchers to visualize the deliberate modulation of the metabolism of cisplatin in human plasma by D-methionine [127], N-acetyl-L-cysteine [128] and sodium thiosulfate [129], which revealed that these SMW sulfur compounds can be used to 'neutralize' a highly toxic cisplatin hydrolysis product that is largely responsible for the severe side-effects of this Pt drug [130]. Thus, metallomics tools can be employed to obtain insight into the time-dependent in vitro metabolism of novel metal complexes in blood plasma, which is useful not only to estimate how much of the administered anticancer metal complex is likely to reach the cancer tissue intact [12,63], but also to gain insight into the formation of degradation products that may cause undesirable severe side-effects at the organ level [130,131].

### 6.2. Metallodrugs-/Novel-Metal(loid)-Complexes-Induced Rupture of RBCs

A novel-metal-entity-induced rupture of RBCs in the bloodstream results in the release of Hb into plasma, which can cause severe toxic side-effects, such as potentially life-threatening thrombotic effects and kidney toxicity [91]. Cisplatin, for example, is well-known to perturb the integrity of cell membranes—resulting in the damage of chicken RBCs [62]—which likely contributes to its severe side-effects in treated patients [56,123]. Based on the well-established metabolism of cisplatin in blood plasma [76], however, it is unknown if the integrity of the RBC lipid bilayer membrane is perturbed by cisplatin or its hydrolysis products. A recently developed metallomics tool can be employed to analyze blood plasma (50 µL) for a hemoglobin–haptoglobin complex to estimate the RBC rupture that is induced by a pharmacologically relevant dose of a metallodrug when added to whole blood [132].

### 6.3. Assessment of the Selectivity of Metallodrugs/Novel Metal(loid) Complexes to Cancer Cells

The selectivity of a novel metal(loid) complex toward the intended cancer can be assessed in cell-culture studies by measuring the $IC_{50}$ value in cancer cells and healthy cells (e.g., fibroblasts). The advent of single cell (sc) ICP-MS over the last couple of years has demonstrated that it can—at least in principle—be employed to observe the uptake of cisplatin into sensitive and resistant cancer cells [133]. Remarkably, this inherent capability offers the ability to probe the selectivity of novel metal(loid) complexes under near-physiological conditions in vitro, with the caveat that, in the presence of the bloodstream, the same selectivity may not be obtainable in vivo.

Faced with the urgent need to accelerate more novel metal(loid) complexes that exhibit the desired biological activity to clinical studies, the application of certain metallomics tools (SEC-ICP-AES) can provide useful information about potentially adverse effects of a metallodrug in the bloodstream (i.e., RBC rupture; degradation in plasma), while other metallomics tools (scICP-MS) can allow for probing the selectivity of the metallodrug to preferentially 'hit' cancer cells compared to healthy cells in vitro [133]. The analysis of a

set of novel metal(loid) complexes with different metallomics tools, therefore, offers the prospect of identifying those that are stable in plasma, do not compromise the integrity of RBCs (in blood) and exhibit selectivity toward cancer cells (Table 3). The complexes that fulfill these criteria can then be forwarded to preclinical and clinical studies. While knowledge about the mechanism of action of an anticancer drug at the biomolecular level is unquestionably useful [134], it is not necessarily a priority when screening novel metal complexes. Instead, their selective delivery to the tumor tissue should be the primary focus [63,135].

## 7. Integrative Metallomics Studies

In order to make progress in terms of linking 'molecular data' of potentially toxic metal(loid) species and cytotoxic metallodrugs with undesirable or desirable human health effects, it is necessary to establish the complete sequence of *bioinorganic chemical reactions* in various *biological compartments* of the bloodstream–organ system (i.e., plasma, RBCs and organs) (Figures 3 and 4) (Table 3). With regard to the exposure of an organism to a toxic metal(loid) species, one potential challenge is to measure an outcome in a target organ (e.g., nutrient deficiency) [69,86,108], while it is comparatively easy to observe the shrinkage/remission of a tumor after treatment with a metallodrug [136]. Our limited understanding of the 'molecular toxicology' of toxic metal(loid) species and metallodrugs in the bloodstream must be attributed to the fact that most studies are conducted using either blood plasma or RBC lysate for practical reasons, since the shelf life of whole blood is rather limited. This approach to addressing the bioinorganic chemistry in individual biological compartments can intrinsically not answer the most crucial question of which specific metal species will impinge on a toxicological target organ and/or tumor in the whole organism. In a sense, this undesirable situation is somewhat reminiscent of the problem that proteomics researchers faced who chose to study purified proteins in isolation for many years without realizing that addressing the biological complexity requires one to study protein–protein interactions to describe the events that unfold inside any given cell [137]. Since it is ultimately the flux of toxic metal(loid) substances that impinge on toxicological target organs/tumors and determine the damage [detrimental in the context of toxic metal(loid) species, but desirable in the context of metallodrugs], a better understanding of the corresponding bioinorganic chemistry in the bloodstream will contribute to advancing toxicology [108], ecotoxicology [138] and pharmacology [97]. Integrating the bioinorganic mechanisms that unfold in the bloodstream with biomolecular uptake mechanisms that are located at the organ surface [139] will inevitably advance human health, as these processes play a crucial role in terms of a) causally linking human exposure to toxic metal(loids) to disease and b) ultimately achieving 'precision oncology' [120]. Some metallomics tools, such as X-ray-based spectroscopy, even offer the capability to probe deeper inside the organ/tumor tissue to obtain information on the sub-cellular compartment that a specific toxic metal(loid) or metallodrug targets in a healthy cell or cancer cell [140]. The innovative applications of metallomics tools thus represent an essential first step in terms of unravelling the potential pharmacology and the mechanism of action of novel metal(loid) complexes to further advance the important role that integrated metallomics are destined to play in health and disease [15].

**Table 3.** Overview of studies which exemplify the usefulness of different metallomics tools to probe the interaction of toxic metal(loid) substances with different biological fluids and/or cells.

| Toxic Metal(loid) Species in Biological Fluid | Metallomics Tool(s) | Investigated Species | Obtained Information | Reference |
|---|---|---|---|---|
| Blood plasma (in vitro) | SEC-ICP-AES | $Cd^{2+}$ | $Cd^{2+}$-driven displacement of $Zn^{2+}$ from a Zn metalloprotein | [16] |
| | SEC-ICP-AES | $CH_3Hg^+$ | formation of hCys-$CH_3Hg^+$ complexes | [84] |
| Red blood cell cytosol (in vitro) | SEC-ICP-AES and XAS | $Hg^{2+}$, $CH_3Hg^+$, $CH_3CH_2Hg^+$, $Cd^{2+}$ | formation of stable complexes of $Hg^{2+}$, $CH_3Hg^+$ and $CH_3CH_2Hg^+$ with hemoglobin | [75] |
| | SEC-ICP-AES and XAS | $CH_3Hg^+$ and $(GS)_2AsSe^-$ | formation of $(GS)_2AsSe$-$HgCH_3$ | [90] |
| Bile (in vivo) | XAS | injection of hamsters with $As^{III}$ and $Se^{IV}$ | detection of $(GS)_2AsSe^-$ | [81] |

| Metallodrug in Biological Fluid/Compartment | Metallomics Tool | Investigated Metallodrug | Obtained Information | Reference |
|---|---|---|---|---|
| Blood plasma (in vitro) | SEC-ICP-AES | cisplatin and carboplatin, Titanocref | stability/degree of hydrolysis/degradation | [76] [63] |
| | SEC-ICP-AES | cisplatin | formation of complexes between a cisplatin-derived hydrolysis product and thiosulfate | [129] |
| | SEC-ICP-AES | cisplatin | formation of complex between a cisplatin-derived hydrolysis product and N-acetyl-L-cysteine | [128] |
| | SEC-ICP-AES | cisplatin | formation of complexes between a cisplatin-derived hydrolysis product and D-methionine | [127] |
| | SEC-ICP-AES | cisplatin | modulation of the metabolism of cisplatin in serum of cancer patients with human serum albumin (HSA) | [141] |
| | SEC-ICP-AES | (2,2′:6′2″-terpyridine) platinum (II) complexes | binding to rabbit serum albumin | [131] |
| | SEC-ICP-AES | cisplatin and NAMI-A | comparative metabolism | [142] |
| Whole blood (in vitro) | SEC-ICP-AES SEC-GFAAS | no metallodrugs investigated | dose-dependent effect of metallodrug on RBC lysis | [99] [132] |
| Healthy and cancer cells (cell culture) | scICP-MS | cisplatin | selectivity of metallodrug | [133] |

## 8. Conclusions

Throughout life, the dynamic flow of toxic metal(loid) species inevitably connects every mammalian organism to the surface geochemistry of our home planet. Because of the paucity of information about possible mechanistic links that functionally connect environmental exposure to toxic metal(loid)s with adverse pregnancy outcomes [143], neurodevelopment in children [144], adverse effects on organs [145] and human diseases of unknown etiology [54,108], gaining insight into the underlying bioinorganic chemistry

in the environment–bloodstream–organ system [146] is critical, especially given the rapidly growing problem of dealing with electronic waste—of which we generate 53.5 million metric t per year [147]. Elucidating this bioinorganic chemistry not only holds the prospect of causally linking human environmental exposure to toxic metal(loid) species with the etiology of neurodegenerative diseases including Alzheimer's Disease [148] and to curb emissions into the environment, but also to develop affordable, practical solutions [149]. Relatedly, the development of better anticancer metal-base drugs hinges on improving our screening process of novel metal complexes to include events that unfold in the bloodstream and, in turn, advance more drug candidates to preclinical studies [12]. The integration of bioinorganic chemistry events in the blood plasma–RBC–organ system is therefore critical to causally link human exposure to toxic metal(loid) species with diseases [150] and to help discover the next generation of metallodrugs.

**Funding:** Funding for some of the studies cited in the article was provided by the Natural Sciences and Engineering Research Council (NSERC) of Canada.

**Institutional Review Board Statement:** Studies that were conducted at the University of Calgary were conducted in accordance with the Declaration of Helsinki, and approved by the Life & Environmental Sciences Animal Care Committee (LESACC) and the Conjoint Health Research Ethics Board (CHREB).

**Informed Consent Statement:** Informed consent was obtained from all subjects that were involved in studies conducted at the University of Calgary.

**Data Availability Statement:** The data presented in this study that were obtained at the University of Calgary are available on request from the corresponding author.

**Acknowledgments:** Maryam Doroudian is supported by a grant from the Natural Sciences and Engineering Research Council (NSERC) of Canada. We greatly appreciate one particular reviewer's feedback which helped to improve the overall quality and clarity of our manuscript.

**Conflicts of Interest:** The authors declare no conflict of interest.

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
