# Peer review of "Integrative Metallomics Studies of Toxic Metal(loid) Substances at the Blood Plasma–Red Blood Cell–Organ/Tumor Nexus"

_inorganics, doi:10.3390/inorganics10110200_

Round 1
Reviewer 1 Report (Previous Reviewer 1)
Accept
Author Response
The reviewer accepted our manuscript, but requested moderate English changes. We thank the reviewer for providing us with feedback and have improved the English style as well as the clarity of the manuscript (see text highlighted in red in revised manuscript 3).
Reviewer 2 Report (Previous Reviewer 2)
Unfortunately, after checking this second version, I feel that the manuscript additions are still very minimal after two revisions. The authors mention that the length has changed from 26 to 30 pages, but the revised manuscript I received still has 16 pages (as the original), and no major content changes/reestructure (just some minor additions (few sentences), and rewrites).
I think that the last two sections are valuable topics, but they really need to be completed and expanded. The perspective should be constructed around them (and not with 1-4 sections being the main part). At the end, sections 5-6 are the main focus -and title- of the perspective. They should include further discussions (including expanding examples and more guidance to novel readers that want to use metalloproteomics to improve clinical translation of metallodrugs -where to start, how to implement, is blood stream the main issue? shall it be a routinary assessment in preclinical studies of novel metal-based compounds?), also linking to Figures 3 and 4 content. There is little use of figures in text, no discussion, and why these schematics are useful.
Regarding figures and tables, despite some figures might be original, I hope the authors agree that the similarity with previously reported figures is extremely high (i.e., significant overlapment that needs to be addressed and adequately cited - at least mentioning that has been adapted from reference...). Additionally if Table 2 is exactly the same, it cannot be used in another manuscript unless explicitely cited (and with the corresponding copyright permissions).
Additional comments:
a) In section 5: The authors mention that metallodrugs are conceptually better suited to target disease-relevant proteins than 1D and 2D organic molecules? Why is that the reason? They do not discuss/show this (only cite one reference), and I am not entirely sure about such a statement. Are there no organic-based drugs approved targeting disease-relevant proteins? Would be surprised... many inhibitors of proteins/enzymes are for indeed organic-based.
b) In section 5 the authors talk about developing smart metallodrugs. What do they understand as smart? And how would metalloproteomics help in this design?
c) According to them, three main problems are associated with metal entities in blood stream (page 9). However, organic-based drugs are also interacting with proteins and can also show stability and also show low tumor selectivity. Why are these unique of metals? Are they sure these are the only challenges? What about drug resistance? What about biodistribution related to speciation? This section would need expansion. Why metalloproteomics will help metallodrugs here? Understanding the mechanisms is crucial, but the perspective should show how (practically and realistically) this can be implemented to truly improve preclinical-to-clinical translation of metallodrugs. The idea is to inspire scientists to consider those tools for preclinical evaluation, and this is not discussed.
d) Building upon previous comment, for instance, they mention only metal complexes do not compromise integrity of RBCs in blood are stable in plasma and selective toward cancer cells should be accelerated to preclinical and clinical studies. How can we know about the selectivity before assessing any preclinical study? How can metalloproteomics assess no off-target accumulation? This is highly unclear to me.
e) Many of the new text contains several typos: e.g., page 3, line 81 (of track changes file): unserscore, page 7 line 248: "enormoustransdisciplinary", page 9 line 343: "manynovel", line 359 "will reduces", and several others. Also new sentences/edits have complicated the text in some parts (e.g., lines 358-361 or 351-353).
Author Response
We greatly appreciate the detailed and constructive feedback of this reviewer! The reviewer mentions that ‘the manuscript additions are still very minimal after two revisions’ and that ‘no major content changes/reestructure (just some minor additions (few sentences), and rewrites’ have been implemented. We note that revision 3 of the manuscript has now increased by approximately 3 pages compared to the original manuscript. This reviewer also pointed out that the last two sections (labelled 5-6 in revised manuscript 2 and 6-7 in revised manuscript 3 owing to a labelling error) need to be completed and expanded.
Based on the reviewer’s comments we decided to change the title from ‘metalloproteomics’ to ‘metallomics studies’ in order to more clearly explain that techniques are not the main focus in this perspective, but rather the integration of the bioinorganic in the bloodstream with what is happening in organs/tumors. We believe that the change that we have implemented into revision 3 also addresses the feedback about the imbalance problem that was mentioned. Specifically we point out that
Regarding the issue that Table 1 was used in one of our previous publications in the Journal Molecules (published by MDPI) we attempted to obtain the copyright permission. On the website of the Journal, however, there was no tab to obtain this permission. We therefore changed the caption of the Table as well as some of its content.
With regard to the figures we point out that Figure 1 was modified from our book chapter. We have therefore changed the corresponding figure caption and references the book chapter. In addition, we improved the quality of all Figures.
The reviewer also provided four main areas in which the manuscript and these sections in particular can be improved, labeled a) to e).
In response, to feedback a) we have attempted to better explain the advantage of 3-dimensional drugs compared to 1 and 2-dimensional organic molecules. We also note that we provided not one, but four recent references for the reader who is interested in this particular aspect. In addition, we never claimed that no small organic based drugs have been approved for targeting disease-relevant proteins. We merely state that 3-dimensional metal-based drugs offer more possibilities in terms of modifying their shape to tailor a drug (e.g. to inhibit the active center of an enzyme).
In response to feedback b) we clarified what we meant by the term ‘smart’ metallodrugs. Since the concept of using metal-based drugs as stimulants of the immune system is just emerging it is soon to evaluate in what way the application of metalloproteomics tools may contribute.
In response to feedback c) we intentionally did not mention organic drug molecules which can of course also result in RBC lysis, low tumor selectivity and or the development of drug resistance. We also implemented considerable changes to the paragraph that starts on the bottom of page 16 (in revised manuscript 3) in which we delineate problems that are associated with the interaction of metal-based drugs with constituents of the bloodstream. With regard to the question ‘What about drug resistance?’ we recognize that this is a considerable problem. Our manuscript specifically focusses on the bloodstream, while the development of resistance is a phenomenon that originates in cancer cells. We have also added two sentences in the section on Metallomics tools to point out that since cell ICP-MS has been successfully applied to study the uptake of cisplatin into sensitive and resistant cancer cells.
In response to the question ‘What about biodistribution related to speciation?’ we have lengthened the section in which we describe the comparative metabolism of cisplatin and carboplatin in human plasma (see last two lines on page 17 and first four lines of page 18 in revised manuscript 3).
One other question raised by the reviewer is how metalloproteomics can be implemented to improve preclinical to clinical translation of metallodrugs. Since we changed the term ‘metalloproteomics’ to ‘metallomics studies’ which clarifies that the focus of our manuscript is not on techniques we believe that the manuscript in its present form (revision 3) sufficiently describes the capability of metallomics tools in the context of future metallodrug development (see text from middle of page 17 and first 6 lines on page 18 in revised manuscript 3.
In response to feedback d) we never claim that only metal-complexes can compromise the integrity of red blood cells and therefore cause adverse health effects. The reviewer also asks ‘how can we know about the selectivity (of a novel metal-based drugs) before assessing any preclinical study’. We respond that only novel metal complexes which exhibit the desired anticancer activity (i.e. an IC50 value comparable to that of cisplatin) along with in vitro studies need to be conducted in order to identify the most promising one for its infusion to preclinical studies. If, however, a novel metal complex ruptures red blood cells and/or degrades in the bloodstream, its potential selectivity is only of limited value in the context of developing new drugs. Another comment from the reviewer was ‘how can metalloproteomics assess no off-target accumulation’ of metal complexes. We respond that a metallomics tool is able to provide insight into its degradation in plasma, but that the toxicity of certain degradation products needs to be assessed separately in cell culture studies to evaluate potential off-target effects of the parent metal complex.
In response to e) we have fixed all of the typos and thank the reviewer for pointing them out to us.
We would like to stress again that the focus of our manuscript is on integrating the bioinorganic chemistry of the toxic metal substances that unfolds in the bloodstream with organ/tumor based effects and not on instrumental analytical metalloproteomics tools per se.
Reviewer 3 Report (Previous Reviewer 3)
This revised manuscript makes some attempt to address the concerns of previous peer review, however it is concerning that changes made to this perspective remain fairly minimal (particularly with respect to sections 1-4 which are essentially only grammatical changes). Revisions of sections 5-6 are more pleasing to see. Nonetheless, the balance of this perspective still could be improved, and does not seem to cover as much detail in 'metalloproteomics' as it does covering the background of toxic metal-based species - is this a perspective article of integrative metalloproteomics, or rather a perspective on toxic metal-based species? I am unsure whether a few additional paragraphs in sections 5-6 sufficiently address previous reviewer concerns; particularly given the limited number of new citations of metalloproteomic techniques. Key contributors in this field remain notably absent (e.g. no examples for Blindauer, Harrington, Goenaga-Infante, single example for Koellensperger).
It is important to say that while I recognise this version improves on the previous submission, I am not sure whether these changes truly reflect the details one would be expected for a perspective article.
I would strongly urge the authors to address the imbalance of sections 1-4 vs. sections 5-6. Alternatively, perhaps the authors may re-consider the title of their perspective article to provide a perspective on "toxic metalloid substances" rather than that of "integrative metalloproteomics" since a detailed evaluation (both breath and depth) of current metalloproteomic analytical approaches could still be greatly improved.
Author Response
We thank the reviewer for the feedback and are glad that the changes we have implemented into section 5 were deemed appropriate. Since it was mentioned that our changes to sections 1-4 were rather minimal, we have therefore lengthened section 4 (page 13, line 4-14, see text highlighted in red in revised manuscript). The reviewer requested additional references from Blindauer, Harrington, Goenaga-Infante and Koellensperger. We note that we have already added references from Blindauer and Koellensperger in revised manuscript 2 (references 16 and 17), but have now included references from Harrington (reference 122) and Goenaga-Infante (reference 123) in the revised manuscript 3. We believe that the suggestion to change the title from ‘integrative metalloproteomics’ to ‘toxic metalloid substances’ will not sufficiently capture the overall focus of our manuscript to provide a perspective that stresses the importance of measuring metalloproteins including endogenous ones and those formed from exogenous metals [toxic metal(loid)s and metal-base drugs] in the bloodstream. Based on the reviewer’s comments we decided to change the title from ‘metalloproteomics’ to ‘metallomics studies’ in order to more clearly explain that techniques are not the main focus in this perspective, but rather the integration of the bioinorganic in the bloodstream with what is happening in organs/tumors.
Reviewer 4 Report (New Reviewer)
Inorganic metal ions and complexes play key roles in life science, and show potential applications in medicine to improve human health. It is necessary to investigate the bioinorganic chemistry of metal(loid) species and metalloproteins to analyze their environmental and pharmacological relevance. In this manuscript, foundational properties of AsIII, Cd2+, Hg2+ (CH3Hg+) and cisplatin were summarized and discussed. The manuscript is scientific significance and is helpful for better understanding of their toxicological and physiological relevance. However, some revisions should be considered for the manuscript.
1. In Tables 1 and 2, molecular properties of major metalloproteins which contain Fe, Cu, Zn in human plasma were listed. Some references about possible effect of AsIII, Cd2+, Hg2+ (CH3Hg+) and cisplatin on these proteins or metal proteins related with AsIII, Cd2+, Hg2+ (CH3Hg+) and cisplatin could be added and generally discussed.
2. Some references about interactions and binding of AsIII, Cd2+, Hg2+ (CH3Hg+) and cisplatin with serum albumin or other proteins could be added and discussed.
3. The language and figures could be refined.
Author Response
We thank the reviewer for the valuable feedback. In response to the request to provide additional references about the interactions, bindings and possible effects of toxic metal(loid) species on serum albumin and other metalloproteins we have added an additional paragraph on page 13 (line 4-14, see text highlighted in red in revised manuscript).
Round 2
Reviewer 2 Report (Previous Reviewer 2)
Despite the additional incorporations, I unfortunately still do not see significant improvement on the manuscript (just some additions), and I truly encourage the authors to rethink what they want to transmit, and re-structure and clearly organize the manuscript (e.g., more focusing on the last two sections but splitting and expanding in different sections about stability due to metal complexes hydrolysis (Pt and Ru probably are different in bloodstream), interaction with major protein like HSA (how this can alter biodistribution, in which cases, how to assess), RBCs, how these interactions can be assessed and can predict tumor accumulation/side-effects, how the speciation in blood stream will enable to predict tumor microenvironment interactions with metallodrugs (drug resistance), etc). The authors keep rewriting some sentences and adding few examples based on suggestions but to me there is no substantial meaningful structural and content changes to the perspective, and there is still very little (or no) analysis in the examples they provide, remaining at a shallow level that do not allow the reader to connect and embrace the topic and see the relevance of integrating metallomics into metallodrug research. Overall, as a reader, it is difficult to understand how metallomic tools/concepts can truly promote clinical translation of metallodrugs beyond what is already known.
As stated, sections 6-7 need to be significantly expanded (but also strongly suggested to be re-organized) to provide informative content to the reader in the medicinal (inorganic) chemistry field. They are still too narrow, and encompass less than 4 pages out of 12, which is not even a 50% of the total content (a perspective should be constituted by a majority -90%- of novel content, since sections 1-4 are mostly introduction from previous works and some minor new edits).
Some comments:
1) They emphasize that their focus is not on techniques, but rather on the incorporation of bioinorganic concepts in routine metallodrug research. Unfortunately, after reading the perspective, I am not sure what the authors try to illustrate. E.g.: they claim that understanding the metabolism of metallodrugs in the bloodstream will help assess selectivity to cancer cells. They state (section 6) that unstability will dramatically increase probability of side-effects and decrease tumor selectivity. I do not see how this links to the tumoral selectivity or toxic side-effects of a metallodrug. Small molecules in general do not show high tumor accumulation, and, indeed they are usually rapidly cleared via kidneys. So: would a fully stable small molecule show higher tumor selectivity? Is that the reason why doxorubicin/paclitaxel are approved and ruthenium complexes not (doxorubicin is not fully stable actually and shows quite many side-effects)? And why and how bioinorganic chemistry tools can/should help here? These questions remain still unanswered to me after reading.
a) Related to the previous comment, in line 379, the authors mention that "low temporal stability of a novel metal complex in plasma will likely inevitably decrease...". Likely? Or inevitably?
b) The authors prefer to remove metalloproteomics from the title to avoid focusing on techniques, but metallomic tools also involve techniques, don't they? Not sure I follow the title change.
2) I do not understand their reply about "smart" metallodrugs and the immune response. Organic-based drugs can also trigger immune responses (e.g., doxorubicin). How bioinorganic tools are going to help to develop "smart" (i.e., more selecrtive) metallodrugs and why is that related to immune responses? The authors do not elaborate on that, and the reader won't know how to proceed if wants to design a smart metallodrug using metallomic tools.
a) In the same line, the authors state (line 433): "Metal complexes that do not compromise the integrity of RBCs in blood, are stable in plasma and exhibit selectivity toward cancer cells can then be accelerated to preclinical and clinical studies". How are they going to know if they are selective to cancer cells before assessing them preclinically? (in vitro is also preclinically). Also cisplatin is not stable (hydrolyzes), but 70% of bimetallic titanium are. Why is cisplatin approved and bimetallic titanium ones no?
3) Regarding the addition of ICP-MS to discuss about drug resistance, this is very limited and simplistic. I understand they want to focus on bloodstream interactions, but, did not the authors also want to relate those interactions with the tumor nexus (i.e., tumor microenvironment, stroma, drug resistance, etc)? And, is not ICP-MS just simply assessing metal content (uptake)? How can this technique provide information about the metallic species and how can this enable us to select candidates for in vivo?
4) I truly do not understand why only cisplatin is used as an example of metallodrug (very briefy other metal complexes appeared but disconnected to the rest of the flow and without a follow-up/elaborated explanation). If they want to have a perspective on metallomics for metallodrugs, they should definitely and significantly expand to other metallodrugs at similar level and connect explanations/illustrate clearly why.
5) The authors added some new lines (lines 295-306), about interactions of metal(oid) compounds with proteins. They mention Cd, methylmercury, cisplatin, but do not elaborate on the implications beyond that or why this is relevant (or how are they connected). Why is relevant that methylmercury for instance binds to rabbit serum albumin? How does this can help e.g., translate further metallodrugs into the clinic?
6) Still several typos remain in the manuscript, even same previously mentioned (e.g., unserscore, line 80, enormoustrans disciplinary, line 244), plus others from new sentences (metallopotein, line 304), etc. Also, in lines 44-45 they mention that chemotherapy's father (and the one who synthesized Salvarsan) was Emil Fischer. What about Paul Ehrlich?
Author Response
Despite the additional incorporations, I unfortunately still do not see significant improvement on the manuscript (just some additions), and I truly encourage the authors to rethink what they want to transmit, and re-structure and clearly organize the manuscript (e.g., more focusing on the last two sections but splitting and expanding in different sections about stability due to metal complexes hydrolysis (Pt and Ru probably are different in bloodstream), interaction with major protein like HSA (how this can alter biodistribution, in which cases, how to assess), RBCs, how these interactions can be assessed and can predict tumor accumulation/side-effects, how the speciation in blood stream will enable to predict tumor microenvironment interactions with metallodrugs (drug resistance), etc). The authors keep rewriting some sentences and adding few examples based on suggestions but to me there is no substantial meaningful structural and content changes to the perspective, and there is still very little (or no) analysis in the examples they provide, remaining at a shallow level that do not allow the reader to connect and embrace the topic and see the relevance of integrating metallomics into metallodrug research. Overall, as a reader, it is difficult to understand how metallomic tools/concepts can truly promote clinical translation of metallodrugs beyond what is already known.
As stated, sections 6-7 need to be significantly expanded (but also strongly suggested to be re-organized) to provide informative content to the reader in the medicinal (inorganic) chemistry field. They are still too narrow, and encompass less than 4 pages out of 12, which is not even a 50% of the total content (a perspective should be constituted by a majority -90%- of novel content, since sections 1-4 are mostly introduction from previous works and some minor new edits).
Response: We thank the reviewer for his detailed comments and also agree with the recommendation to re-structure the manuscript to improve its overall clarity. Consequently, we have significantly restructured section 6 into an introductory paragraph followed by sub-headers, which focus on the stability of metal complexes (6.1.), the interaction of metal complexes with red blood cells (6.2.) and the selectivity of metal complexes (6.3.). We have discussed the usefulness of metallomics tools to gain insight into each of these aspects in each section. This reviewer also comments that our manuscript is ‘too narrow’, particularly with regard to the reader in medicinal (inorganic) chemistry. We reiterate that our manuscript specifically attempts to provide a balanced perspective of what we currently know about the fate of a) toxic metal(loid)s and b) metallodrugs in the bloodstream. This focus inherently requires us to present a balanced view of both aspects and does not allow us to go into more depth with regard to medicinal (inorganic) chemistry. Regarding the balance of the main parts we point out that sections 1-4 provide an introduction and a juxtaposition of various aspects that pertain to conceptual similarities and differences between the fate of toxic metal(loid)s and metallodrugs in the bloodstream. To the best of our knowledge this is the first time that this has been done in the peer reviewed literature. In our opinion sections 1-4 are not just an ‘introduction from previous works with some minor new edits’, but rather the first attempt to discuss the underlying similarities and differences of toxic metal(loid)s and metallodrugs in the bloodstream. This is also the reason why we submitted it as a perspective. The last two sections before the conclusion provide some in depth discussion about toxic metal(loid)s (Section 5, 1327 words in R4) and metallodrugs (Section 6, 1428 words in R4). The fact that they are of equal length (~2.5 pages each) underscores that we have achieved a balanced discussion of the behavior of toxic metal(loid)s and metallodrugs in the bloodstream. We believe that restructuring our manuscript (>10,500 words) has not only significantly improved its clarity, but also provides sufficient detail for the reader of Inorganics.
Some comments:
1) They emphasize that their focus is not on techniques, but rather on the incorporation of bioinorganic concepts in routine metallodrug research. Unfortunately, after reading the perspective, I am not sure what the authors try to illustrate. E.g.: they claim that understanding the metabolism of metallodrugs in the bloodstream will help assess selectivity to cancer cells. They state (section 6) that unstability will dramatically increase probability of side-effects and decrease tumor selectivity. I do not see how this links to the tumoral selectivity or toxic side-effects of a metallodrug. Small molecules in general do not show high tumor accumulation, and, indeed they are usually rapidly cleared via kidneys. So: would a fully stable small molecule show higher tumor selectivity? Is that the reason why doxorubicin/paclitaxel are approved and ruthenium complexes not (doxorubicin is not fully stable actually and shows quite many side-effects)? And why and how bioinorganic chemistry tools can/should help here? These questions remain still unanswered to me after reading.
Response: We never claim that ‘understanding the metabolism of metallodrugs in the bloodstream will help assess selectivity to cancer cells’, but do mention that one metallomics technique (specifically ssICP-MS; see 6.3.) has been successfully employed to gain insight into the relative uptake of cisplatin into cisplatin-susceptible and cisplatin-resistant cancer cells, which provides insight into the selectivity of cisplatin in these particular cells. With regard to the statement in section 6 that ‘unstability will dramatically increase side-effects and decrease tumor selectivity’ we have clearly explained that this statement pertains to cisplatin (see paragraph preceeding section 6.3) and reiterate that from a conceptual point of view the degradation of a metal complex - that exhibits selectivity to cancer tissue - in the bloodstream will inevitably result in breakdown products that lack the selectivity of the parent metal complex (i.e. it is not the intact complex, but the degradation products that impinge on the cancer tissue). The degradation of the metal complex will also result in potentially more severe toxic side effects if the breakdown products are more toxic than the parent metal complex, which is certainly the case for cisplatin.
With regard to the reviewers’s question if a ‘fully stable small molecule would show higher tumor selectivity?’
Response: This depends on whether the intact small stable molecule itself exhibits tumor selectivity (i.e. its IC50 is lower in cancer cells compared to that in healthy cells).
This reviewer also raised another question ‘Is that the reason why doxorubicin/paclitaxel are approved and ruthenium complexes not’.
Response: We cannot answer as we are not familiar with the mechanism of action of these anticancer drugs.
Regarding the reviewer’s question ‘And why and how bioinorganic chemistry tools can/should help here?’
Response: We believe that we clearly outline in the manuscript that metallomics tools can detect a degradation of the parent metal complex in blood plasma which – to reiterate what we already stated – will reduce the selectivity of the investigated metallodrug and increase the possibility of severe side effects if the degradation products are more toxic than the parent drug (this is the case for cisplatin). We hope to have adequately answered these questions by the reviewer which is also documented in the revised manuscript.
- a) Related to the previous comment, in line 379, the authors mention that "low temporal stability of a novel metal complex in plasma will likely inevitably decrease...". Likely? Or inevitably?
Response: After we accepted the changes we implemented in revision 3 this sentence was removed because it was redundant.
- b) The authors prefer to remove metalloproteomics from the title to avoid focusing on techniques, but metallomic tools also involve techniques, don't they? Not sure I follow the title change.
Response: We changed the title because some of the metal-containing metabolites that contain toxic metal(loid)s and or metallodrug-degradation products are actually small molecules [e.g. (GS)2AsSe-], cisplatin-hydrolysis products) and not metalloproteins. We believe that the slightly modified title is now better aligned with the content of our manuscript.
2) I do not understand their reply about "smart" metallodrugs and the immune response. Organic-based drugs can also trigger immune responses (e.g., doxorubicin). How bioinorganic tools are going to help to develop "smart" (i.e., more selective) metallodrugs and why is that related to immune responses? The authors do not elaborate on that, and the reader won't know how to proceed if wants to design a smart metallodrug using metallomic tools.
Response: We never claim that metallomics tools help to develop smart metallodrugs, but merely provide an explanation of the inherent advantages that metallodrugs offer compared to small organic molecules. We clearly have modified the sentence corresponding to this statement (see first paragraph of section 6 in R4).
- a) In the same line, the authors state (line 433): "Metal complexes that do not compromise the integrity of RBCs in blood, are stable in plasma and exhibit selectivity toward cancer cells can then be accelerated to preclinical and clinical studies". How are they going to know if they are selective to cancer cells before assessing them preclinically? (in vitro is also preclinically). Also cisplatin is not stable (hydrolyzes), but 70% of bimetallic titanium are. Why is cisplatin approved and bimetallic titanium ones no?
Response: We cannot provide a more detailed explanation about ‘how are going to know if they are selective to cancer cells before assessing them preclinically‘ as this concept is the focus of a grant application, which we cannot disclose because its content is confidential. We point out that information about the selectivity of a drug in vitro can be in principle obtained by executing cell culture studies in which cancer and healthy cells are treated with the drugs and the relative IC50 values are measured. Owing to the complexity of the bloodstream, however, the selectivity of the same drug in vivo may be different. In terms of why cisplatin is FDA approved, but not the bimetallic Ti drug: cisplatin was approved by the FDA in 1978, when nothing was known about its hydrolysis in the bloodstream and that these processes are functionally connected to its severe adverse side effects. Conversely, the bimetallic Ti drug Titanocref was synthesized comparatively recently and it has not gone though all four phases of testing, which takes not only a considerable amount of time, but is also associated with immense costs (it costs 5 billion $ to get a new drug on the market). In addition, the stability of a metal complex is an important, bot not the only factor to assess its pharmacological potential.
3) Regarding the addition of ICP-MS to discuss about drug resistance, this is very limited and simplistic. I understand they want to focus on bloodstream interactions, but, did not the authors also want to relate those interactions with the tumor nexus (i.e., tumor microenvironment, stroma, drug resistance, etc)? And, is not ICP-MS just simply assessing metal content (uptake)? How can this technique provide information about the metallic species and how can this enable us to select candidates for in vivo?
Response: With regard to our discussion of the use of ssICP-MS and as it relates to drug resistance: the paper that we cite is from a group in Spain which has demonstrated a more selective uptake of cisplatin into cisplatin susceptible compared to cisplatin-resistant cancer cells. Based on these results one can similarly envision the possibility to observe the uptake of a novel metal complex into cancer cells vs healthy cells (e.g. fibroblasts), which is more difficult. Since the title of Section 6 is ‘…plasma/RBC/organ/tumor nexus’ we believe that this phrase includes the tumor microenvironment. In addition, we briefly talk about drug resistance when we summarize the results that were obtained by the Spanish group.
4) I truly do not understand why only cisplatin is used as an example of metallodrug (very briefy other metal complexes appeared but disconnected to the rest of the flow and without a follow-up/elaborated explanation). If they want to have a perspective on metallomics for metallodrugs, they should definitely and significantly expand to other metallodrugs at similar level and connect explanations/illustrate clearly why.
Response: We chose to illustrate the importance of interactions of a novel metal complex in the bloodstream using cisplatin because it is the oldest metallodrug about which we know more compared to other metallodrugs. We point out that we also mention oxaliplatin (new sentence in R4), carboplatin as well as Titanocref. Thus, we discuss four potential metallodrugs in section 6 in R4.
5) The authors added some new lines (lines 295-306), about interactions of metal(oid) compounds with proteins. They mention Cd, methylmercury, cisplatin, but do not elaborate on the implications beyond that or why this is relevant (or how are they connected). Why is relevant that methylmercury for instance binds to rabbit serum albumin? How does this can help e.g., translate further metallodrugs into the clinic?
Response: We have added a new sentence in the revised manuscript in which we explain why the binding of methylmercury to rabbit serum albumin matters. In addition, we have added a sentence to explain why the binding of metallodrugs to HSA is of particular relevance in the development of new drugs.
6) Still several typos remain in the manuscript, even same previously mentioned (e.g., unserscore, line 80, enormoustrans disciplinary, line 244), plus others from new sentences (metallopotein, line 304), etc. Also, in lines 44-45 they mention that chemotherapy's father (and the one who synthesized Salvarsan) was Emil Fischer. What about Paul Ehrlich?
Response: We are thankful for these comments and have corrected these in the revised manuscript (R4) including the replacement of Emil Fischer by Paul E
Reviewer 3 Report (Previous Reviewer 3)
The authors have significantly improved the content of the latter sections of this report, providing much greater detail of studies currently underway in this field. As such, this review now presents itself with a greater emphasis on metallomics rather than mainly providing a description of what toxic metal(loid) substances may/do exist. I am satisfied that this manuscript is now suitable for publication in Inorganics.
Author Response
This reviewer stated that the authors have significantly improved the content of the latter sections and that he/she is now satisfied that this manuscript is now suitable for publication in Inorganics.
This manuscript is a resubmission of an earlier submission. The following is a list of the peer review reports and author responses from that submission.
Round 1
Reviewer 1 Report
In the perspective, Doroudian and Gailer provided summary of the bioinorganic chemistry of As3+, Cd2+, Hg2+ and CH3Hg+ and the anticancer metal-based drugs cisplatin in the bloodstream. The review comprehensively pointed out the importance of studying toxic metal(loid) species and metal-based drugs in the bloodstream, well summarized the toxic metal(loid) species and metal-based drugs in blood, the bioinorganic chemistry of toxic metal(loid) substances in the bloodstream in vitro and in vivo, the toxic metal(loid) species at the plasma/RBC/organ nexus, the metal-based drugs/novel metal(loid) complexes at the plasma/RBC/organ/tumor nexus, and briefly mentioned the integrative metalloproteomics. Overall, the review filled in the gap of bioinorganic chemistry of toxic metals with a focus on environmental pollutions and worth a publication in “Inorganics”. The authors used “Integrative Metalloproteomics” as key word for the perspective in the title, however, there is only very brief summary of this part. I would strongly suggest the authors to comprehensively discuss this part since it is an emerging approach to study the toxicology of metals.
Author Response
We greatly appreciate this reviewer’s comments that our review fills a gap to address the bioinorganic chemistry of toxic metals with a focus on environmental pollution that should be published in 'Inorganics'. With regard to the suggestion to more comprehensively discuss the ‘integrative metalloproteomics’ part, we believe that we sufficiently explain what we mean by this term, namely the need to integrate the bioinorganic chemistry of toxic metal(loids)/metal-based drugs in the bloodstream with biomolecular events that unfold within organs. We have, however, slightly improved this paragraph to further improve its clarity (see text highlighted in red in revised manuscript).
Reviewer 2 Report
The perspective submitted by Doroudian and Gailer attempts to show how the integration of metalloproteomic techniques in metal-based anticancer drugs evaluation (together with the increasing understanding on metal(loid) toxicity-disease relationship) can assist in translating more anticancer metal complexes to clinical trials, to overall improve patient´s health and quality of life.
While the topic is definitely interesting and timely, and I do believe that the authors have sufficient background on (metal) toxicology to provide new avenues to the metal-based (anticancer) medicine, I also think that the perspective has not fulfilled its main purpose.
The idea of the perspective can be promising, but requires further development, especially from the tumor and metal anticancer research side (i.e., beyond Pt, discussing how and where to integrate metallomics, how this can help significantly the design of novel complexes, how this could explain failures of metal complexes in clinical trials -Pt and non-Pt, etc). From my point of view, the perspective is too short-sighted, with a high focus on toxicology of metals like mercury, cadmium or methylmercury (almost all discussions/relevant examples are based on them), while little is mentioned about tumor features and metals relevant in anticancer research (As has some comments, but Pt, Ru, Cu, Fe, etc do not dominate the perspective ).
The authors only briefly focus a bit more on this on point 5 (almost last section), where mention how metallomics have contributed to explain side-effects of only cisplatin and carboplatin (approved more than 20 years ago). Sections 1-4 are more an introduction, with many concepts overlapping with previous reviews/perspective of the same author(s) (e.g., T. G. B., P. K. and J. Gailer, Molecules, 2021, 26, 3408). Many sections are very similar (e.g., RBC-organ interface, or table 1 in both perspective: molecules 2021 and the submitted one), and even with some some identical wordings in several sections.
Overall, I would recommend the authors to rethink and restructure the perspective (the topic can be interesting), but shortening first sections and building upon 5 and 6 (whose title indeed strongly links with the title of the perspective) significantly towards Pt- and other recent metal anticancer.
As suggestion, and after reading the perspective, several questions still remain unanswered, e.g.:
- How these proteomic tools can help to really solve side-effect issues in metal-based drugs? Usually metal-based drugs that are highly stable do not display high reactivity, while high reactivity do usually entail also undesired interactions, but still may work (e.g., cisplatin).
- Proteins and RBCs can act as carriers for anticancer compounds. How can this be taken into account when assessing interactions? What about the immune system in this regard?
- How does the metal nature influence in this study? Beyond Pt, is there any other metal explored in anticancer research which has benefited (and/or can) from these (metalloproteomic) tools?
- Pt is exogenous, so can fall in similar category of As, Cd, etc. But what about Fe, Cu, Zn, i.e., physiological metals that are essentials and can be also employed in (anticancer) therapy?
- Which tools/techniques can we concretely use to study this (for metal anticancer) and how should we integrate them in the design of future metallodrugs?
- The authors claim that the low number of metal-based anticancer compounds in the clinic can be solved by assessing effect on RBCs, blood plasma stability and selectivity between cancer versus healthy cells. I believe this is quite a simplistic and general statement (many factors hamper clinical translation of metallodrugs), and that they should elaborate more on this. E.g., cisplatin/carboplatin hydrolysis (instability) in front of certain biomolecules is indeed one of the reasons of its efficacy (and also of its side-effects), while for oxaliplatin might not be that needed.
Author Response
We greatly appreciate this reviewer’s detailed comments on our manuscript.
- The reviewer thinks that our perspective has not fulfilled its main purpose. This opinion is in contrast to the feedback from reviewer 1 and 3.
- The reviewer thinks that our perspective is ‘too shortsighted’ and ‘especially from the tumor and metal anticancer research side’, thinks that there is a ‘high focus on toxicology of metals like mercury’ and that our perspective does not address questions as to ‘how and where to integrate metallomics, how this can help significantly the design of novel complexes, how this could explain failures of metal complexes in clinical trials (-Pt and non-Pt)’. We believe that our perspective equally focuses on toxic metal(loid)s from the environment and metal-based drugs and provides a big picture of why integrative metalloproteomics has gained in importance as an emerging approach in this field. With regard to the ‘how and where to integrate metallomics’ we specifically address this very aspect in ‘6. Integrative metalloprotoeomics’. In terms of addressing ‘how this can help to explain failures of metal complexes’ we have included additional sentences on page 15 (see text highlighted in red in the revised manuscript) in which we specifically identify the problem that in vitro studies of cytotoxic metal complexes sometimes fail to result in a desired effect in vivo.
- The reviewer mentions that no information about tumor features and metals relevant in anticancer research is provided. Owing to the focus of our perspective we cannot provide details about ‘tumor features’. Furthermore, we chose cisplatin to illustrate the problem associated with the translation of more novel metal-complexes to successful metal-based drugs, namely the lack of understanding their metabolism/degradation in the bloodstream.
- This reviewer states that ‘Sections 1-4 are more an introduction, with many concepts overlapping with previous reviews/perspective of the same author(s)’ with ‘even some identical wordings in several sections’. While we recycled some formulations from our ‘Molecules’ article, the current article is original in the sense that it is the first of its kind to discuss both toxic metal species - environmentally abundant toxic metal(loid)s and metal-based ones together.
- This reviewer suggests to ‘shortening first sections and building upon 5 and 6’. We believe that sections 1-4 are crucial to convey our perspective to the reader. However, we have partially lengthened 5 and 6 in the context of addressing the other reviewer’s feedback.
- This reviewer wanted us to address several remaining questions, such as ‘How these proteomic tools can help to really solve side-effect issues in metal-based drugs?’ On page 16 we specifically state that: ‘ Furthermore, the application of metallomics tools has shown to play a pivotal role in terms of potentially reducing the severe toxic side effects of cisplatin by the co-administration of small molecular weight molecules to human plasma to neutralize a highly toxic hydrolysis product that is likely responsible for its severe toxic side-effects.’.
- With regard to the reviewer’s questions that ‘Proteins and RBCs can act as carriers for anticancer compounds. How can this be taken into account when assessing interactions? What about the immune system?’ Although we agree that these are important considerations, we believe that they are beyond the scope of this perspective.
- With regard to the comment that ‘Beyond Pt, is there any other metal explored in anticancer research which has benefited (and/or can) from these (metalloproteomic) tools’ we have added an additional sentence into the revised manuscript that refers to a study in which the stability of a titanium-gold-containing complex was investigated in blood plasma (see sentence highlighted in red on page 16 of the revised manuscript).
- The reviewer also asks ‘But what about Fe, Cu, Zn, i.e., physiological metals that are essential and can be also employed in (anticancer) therapy?’. We thank the reviewer for this suggestion, but think that addressing this question is beyond this perspective's scope as we focus on toxic metal(loid)s and metal-based drugs rather than essential elements as anticancer drugs.
- With regard to the question ‘Which tools/techniques can we concretely use to study this (for metal anticancer) and how should we integrate them in the design of future metallodrugs?’, we that the reviewer for pointing this out. We respond that our perspective is not intended to discuss specific tools and we refer the interested reader to relevant reviews on page 4 (refs 13-17).
- The reviewer also states that ‘The authors claim that the low number of metal-based anticancer compounds in the clinic can be solved by assessing effect on RBCs, blood plasma stability and selectivity between cancer versus healthy cells. I believe this is quite a simplistic and general statement (many factors hamper clinical translation of metallodrugs), and that they should elaborate more on this.’. While we agree that many other factors hamper the clinical translation, all metal-based anticancer drugs are intravenously administered emphasizing the tremendous importance of their bioinorganic chemistry in the bloodstream in terms of achieving the desired pharmacological outcome. We therefore decided to focus on the factors that fall into our expertise.
Reviewer 3 Report
This article from Doroudian and Gailer provides an interesting perspective on the biochemical interactions of toxic metal(loid) substances ranging from point of contact (inhalation, ingestion, injection) to extracellular and intracellular interactions. I find this review to be well written and will be particularly beneficial for those who are somewhat unfamiliar with the field. My only recommendations to further improve the quality of this work would be for the authors to consider a discussion of specific approaches which are employed to study metalloproteomics in the systems described, and to consider the challenges that we are still up against in the bio-inorganic community - there appears to be little mention of techniques which are employed (beyond a brief mention of HPLC and 2D-PAGE). The references include some examples of hyphenated analysis techniques (LC-ICP) though I felt this discussion was somewhat lacking in a perspectives concerned with metalloproteomics, with contributions from important contributors in the field notably absent (e.g. Blindauer, Koellensperger). While I acknowledge that this perspective provides a more general overview of the area, rather than a perspective on analytical chemistry, I believe some discussion of practical techniques and limitations would greatly enhance the quality and impact of this work.
Author Response
We greatly appreciate this reviewer’s feedback and want to point out that we submitted our manuscript as a perspective that aims to provide a big picture about recent progress in the area of integrative metalloproteomics. Therefore, we do not think that details about the techniques are needed and we specifically refer the reader to relevant reviews which describe state of the art analytical tools that are being used in metalloproteomics research. We did, however, include two important contributions from Blindauer and Koellensperger which we failed to include into the original manuscript. Please see highlighted references 16-17 in the revised manuscript (page 4) and the corresponding references highlighted in red.
Round 2
Reviewer 2 Report
I thank the authors for the extensive reply to my comments. Unfortunately, almost none of these have been addressed, and the changes made to the perspective are really minimal. In my opinion, the others reviewers were also strongly advising to expand metalloproteomics sections (i.e., the last sections).
I agree that, in a review, recycling some figures (with the appropiate citation in the caption -which the authors do not have in any of the cases-), might be valuable in certain cases (not that much in a perspective, and even more unusal with tables), but the proportion of novel information in any new manuscript should always be drastically higher than the re-used/recycled parts. This is not the case here, and I still believe that if the authors really want to provide a perspective in the field, they should significantly expand sections 5 and 6 (more than one sentence), including details on techniques, discussion about other metals (to note, Cu and Fe complexes are also evaluated as metal-based drugs) and a bit more on how metal-tumor-related aspects might play a role in these analyses (especially if they claim in the abstract that this integrative approach will help to improve clinical translation of metal-based anticancer drugs).